# Emodin as an Inhibitor of PRV Infection In Vitro and In Vivo

**DOI:** 10.3390/molecules28186567

**Published:** 2023-09-11

**Authors:** Xiaojing Cai, Zhiying Wang, Xiaocheng Li, Jing Zhang, Zhiyuan Ren, Yi Shao, Yongkang Xu, Yan Zhu

**Affiliations:** 1College of Veterinary Medicine, Northeast Agricultural University, Harbin 150030, China; caixiaojing777@126.com (X.C.); wzyzy0419@163.com (Z.W.); zhiyuanren0328@163.com (Z.R.); sy15049522319@163.com (Y.S.); 18754963560@163.com (Y.X.); 2Harbin Da BEINONG Animal Husbandry Technology Co., Ltd., Harbin 150030, China; lxch_215@yeah.net (X.L.); zhangjingdbn1314@126.com (J.Z.)

**Keywords:** antiviral, emodin, pseudorabies virus

## Abstract

Pseudorabies (PR) is an acute and severe infectious disease caused by pseudorabies virus (PRV). Once the virus infects pigs, it is difficult to eliminate, resulting in major economic losses to the global pig industry. In addition, reports of human infection with PRV suggest that the virus is a potential threat to human health; thus, its significance to public health should be considered. In this paper, the anti-PRV activities of emodin in vitro and in vivo, and its mechanism of action were studied. The results showed that emodin inhibited the proliferation of PRV in PK15 cells in a dose-dependent manner, with an IC50 of 0.127 mg/mL and a selection index of 5.52. The addition of emodin at different stages of viral infection showed that emodin inhibited intracellular replication. Emodin significantly inhibited the expression of the IE180, EP0, UL29, UL44, US6, and UL27 genes of PRV within 48 h. Emodin also significantly inhibited the expression of PRV gB and gD proteins. The molecular docking results suggested that emodin might form hydrogen bonds with PRV gB and gD proteins and affect the structure of viral proteins. Emodin effectively inhibited the apoptosis induced by PRV infection. Moreover, emodin showed a good protective effect on PRV-infected mice. During the experimental period, all the control PRV-infected mice died resulting in a survival rate of 0%, while the survival rate of emodin-treated mice was 28.5%. Emodin also significantly inhibited the replication of PRV in the heart, liver, brain, kidneys and lungs of mice and alleviated tissue and organ damage caused by PRV infection. Emodin was able to combat viral infection by regulating the levels of the cytokines TNF-α, IFN-γ, IL-6, and IL-4 in the sera of infected mice. These results indicate that emodin has good anti-PRV activity in vitro and in vivo, and is expected to be a new agent for the prevention and control of PRV infection.

## 1. Introduction

Pseudorabies (PR) is an acute and highly infectious disease caused by pseudorabies virus (PRV) that has caused major economic losses to the global pig industry. PRV belongs to the Herpesviridae family, a member of the Alphaherpesvirinae family which has a broad spectrum of infected hosts and can cause clear cytopathic changes in infected cells. It is an important model virus for exploring the biological characteristics of herpesviruses [1]. The clinical manifestations in pigs, the only natural host, vary after infection from subclinical symptoms to death; however, PRV infection is usually fatal in newborn piglets and susceptible animals of unnatural host species, and animals eventually die due to fatal neurotoxicity [2]. In recent years, attenuated vaccines with deleted virulence genes and matching serological diagnostic tests have been used to try to control PRV infection in pigs worldwide. However, due to long-term immune pressure, new PRV variant strains continue to be detected in the clinic, resulting in new challenges for the prevention and control of PR [3,4]. In addition, PRV has been found to infect humans and cause central nervous system (CNS) diseases, fatal encephalitis and endophthalmitis [5,6,7], suggesting that PRV infection has incurred new and potential future public health hazards. Therefore, the development of safe and effective drugs against PRV infection is a beneficial and necessary addition to reduce the PRV infection rate in pigs and eliminate its public health threat.

Natural products have significant advantages, such as long action times and low toxicity, and the virus itself does not easily develop drug resistance, which is beneficial for the development of new antiviral drugs [8,9]. Regarding new drugs, plant-derived natural products are the main source of the discovery of new therapeutic agents due to their chemical and structural diversity [10]. Many natural products of plant origin have been proven safe and effective for the treatment of PRV infection. Resveratrol is a stilbene compound extracted from the leaves of grapevines. Studies have found that it can inhibit the decline in cell viability and the production of viral DNA caused by PRV infection [11], while also effectively inhibiting the virus propagation of PRV in infected piglets, alleviating PRV-induced inflammation and enhancing the immunity of infected piglets [12]. Quercetin is a natural compound that exists in a wide variety of Chinese herbal medicines. It shows significant antiviral activity against a variety of PRV strains, can effectively prevent the adsorption and entry of PRV infection, and can protect PRV-infected mice from lethal attack [13]. Luteolin is an important component of honeysuckle and Perilla frutescens. It can inhibit the replication of PRV in PK-15 cells by inhibiting gB gene and protein expression and by alleviating virus-induced apoptosis. Moreover, it can play a protective role in PRV-infected mice by delaying symptom onset time, improving survival rate, reducing viral load in tissues and organs, and improving the inflammatory response [14].

Emodin is a natural anthraquinone derivative that exists as an active ingredient in rhubarb, Polygonum cuspidatum, and P. multiflorum [15]. Emodin has a wide range of pharmacological properties, including antiviral, antibacterial, anti-inflammatory, antitumor, immunosuppressive, and neuroprotective properties [16]. Especially in terms of antiviral properties, it has been shown to have a variety of antiviral activities, including against herpes simplex virus, SARS coronavirus, influenza A virus and porcine reproductive and respiratory syndrome virus [17,18,19,20]. Emodin has problems such as poor water solubility, low biocompatibility, rapid systemic elimination, and off-target side effects, which lead to an unsatisfactory therapeutic effect [21]. Despite these important advances, the antiviral activity of emodin against PRV has not been studied. Therefore, this study evaluated the antiviral effect of emodin on PRV infection in vivo and in vitro, and explored its mechanism of action against PRV in an endeavor to find a new drug to prevent PRV infection.

## 2. Results

### 2.1. Emodin Inhibits the Proliferation of PRV in PK-15 Cells

In this study, DMEM was used to dilute emodin, and the final concentration of DMSO was less than 0.5%, thus having no effect on viruses and cells. The cytotoxicity of emodin was determined by microscopic examination and a CCK-8 assay to evaluate PK-15 cell viability. The results showed that the cell viability gradually decreased with increasing emodin concentration. When the concentration of emodin was less than 0.25 mg/mL, the viability of PK-15 cells was not significantly different from that of the cell control group (Figure 1B); thus, this concentration was selected as a safe concentration for subsequent cell experiments. The CC50 value of emodin was calculated to be 0.7005 mg/mL. The inhibitory activity of emodin on PRV in PK-15 cells was evaluated by the CCK-8 method and FQ-PCR. The CCK-8 results showed that emodin significantly inhibited the cell death induced by PRV infection in a dose-dependent manner. When the concentration of emodin was 0.25 mg/mL, the inhibition rate was as high as 96.35%, and when the concentration of emodin was 0.125 mg/mL, the inhibition rate of PRV was still greater than 50% (Figure 1C). The EC50 of emodin was calculated to be 0.127 mg/mL, and the SI was 5.52. In the FQ-PCR assay, the virus copy number decreased in a dose-dependent manner in the presence of emodin. When the concentration of emodin was 0.25 mg/mL, 0.125 mg/mL and 0.0625 mg/mL, the virus copy number decreased by 31.71-, 14.06- and 3.29-fold, respectively (Figure 1D). In conclusion, emodin can significantly inhibit the proliferation of PRV in PK-15 cells.

### 2.2. Effect of Emodin on Stages of the PRV Life Cycle

To further explore the antiviral mechanism of emodin in PK-15 cells, FQ-PCR was used to determine the viral copy number of infected cells at different stages of the PRV life cycle. In the cell pretreatment assay, cells were first pretreated with emodin and then infected with PRV, but no anti-PRV effect was detected (Figure 2A). In the inactivation assay, emodin and PRV were pretreated at 37 °C for 1 h and then added to the cells for coincubation. The results showed that there was no significant difference in the virus copy number between the experimental group treated with emodin and the virus control group without emodin, indicating that emodin could not directly inactivate PRV virions (Figure 2B). The effect of emodin was not obvious in the adsorption test (Figure 2C) and entry test (Figure 2D). Once inside the cell, the virus begins to replicate itself using the raw materials and energy systems provided by the host cell. In the virus replication assay, emodin significantly inhibited PRV replication (Figure 2E).

### 2.3. Emodin Inhibits PRV Gene Expression

PRV gene expression is the first step after viral DNA enters the nucleus, and gene transcription is a key step in viral replication. The expression of the PRV immediate early gene IE180, early gene EP0 and PRV infection-related genes UL29, US6, UL27 and UL44 were detected by qPCR at 3, 6, 12, 24 and 48 hpi. The relative expression levels of the tested genes are shown in Figure 3. The expression levels of all tested genes in the emodin treatment group showed a trend to increase within 48 h; that is, the mRNA expression levels of the tested genes increased with time. However, compared with the virus control group, emodin still significantly inhibited the expression of all tested genes within 48 h (*p* < 0.05).

### 2.4. Effect of Emodin on PRV gB and gD Protein Expression

The gB protein is one of the most conserved envelope glycoproteins in the herpesvirus family and is essential for the fusion process of virus entry and cell-to-cell transmission [22]. The gD protein is an important cofactor in the PRV entry process and is responsible for recognizing the virus and binding it to specific cellular receptors [23]. To further study the effect of emodin on the expression of the PRV gB protein and gD protein, the expression levels of the PRV gB protein and gD protein in PRV-infected cells were detected by Western blotting, and the target sites of emodin binding to the PRV gB protein and gD protein were predicted by molecular docking. We found that emodin could significantly inhibit the expression of PRV gB protein and gD protein. In addition, emodin can form hydrogen bond interactions with the GLU-141 amino acid site of the PRV gB protein (Figure 4E), and the binding energy is −5.66 kcal/mol. It can also form hydrogen bond interactions with the PRV gD protein at the ASP-64, GLU-63 and ALA-248 residues (Figure 4F), and the binding energy is −5.25 kcal/mol. Emodin could stably bind to PRV gB protein and gD protein, thereby affecting the expression of these proteins and inhibiting viral replication.

### 2.5. Emodin Alleviates PRV-Induced Apoptosis

Viral infection usually leads to apoptosis. To evaluate the effect of emodin on the apoptosis of infected cells, PRV-infected cells were treated with emodin, and apoptosis detection was performed by flow cytometry. The results of the apoptosis assay showed that 36 h after virus infection, the apoptosis rate of the normal control group was 7.2% (Figure 5A), that of the virus control group was 14% (Figure 5B) and that of infected cells treated with emodin at a concentration of 0.25 mg/mL was 8.8% (Figure 5C). These results showed that PRV infection induced PK-15 cell apoptosis, while emodin inhibited the apoptosis induced by PRV infection.

### 2.6. Emodin Inhibits PRV Infection in Mice

Previous experimental results confirmed that emodin has great anti-PRV activity in vitro, and to verify its inhibitory effect in vivo, animal experiments were performed as described above. On the third day after challenge, the mice in the virus control group began to exhibit progressive neurological symptoms, including abnormal excitement, biting of the injection site, skin damage at the injection site, and bleeding. Some mice were sacrificed on the third day after the PRV challenge, and the remainder died on the fifth day, with a survival rate of 0%. In the emodin administration group, mice developed clinical symptoms and died on the fourth day after challenge, but two mice were still surviving on the seventh day, for a survival rate of 28.5%. All mice in the normal control group survived (Figure 6A). The weight change curve of mice showed that compared to the normal control group, the weight of mice in the virus control group continuously decreased after symptoms appeared on the third day. The average body weight loss of mice in the emodin administration group slowed on the 4th and 5th days, and showed a significant upward trend after the 6th day (Figure 6B).

To determine the viral load of mouse tissues and organs, three mice were randomly selected from each group; heart, liver, lung, kidney and brain tissues were collected and weighed; the total DNA was extracted from tissues and organs; and the viral DNA copy number changes were determined by FQ-PCR. The results showed that in the virus control group, PRV could be detected in the heart, liver, lung, kidney and brain tissues of mice, and the viral load in brain tissues was the highest, followed by liver, lung, kidney and heart. Compared to the virus control group, the viral load in the brain, liver and lung tissues of mice in the emodin administration group was significantly reduced (*p* < 0.05), especially in the brain tissue, and the viral load in the brains of infected mice after emodin administration was reduced by approximately 4.01-fold (Figure 6C).

To further explore whether emodin has a protective effect on the tissue damage caused by PRV, histopathological examination was performed on the heart, liver, lung, kidney and brain tissues of mice in each group. As shown in Figure 6D, the HE section results showed that the hearts of the mice in the virus control group showed myocardial fiber breakage, disordered myocardial fiber arrangement and slight bleeding, extensive hepatocyte edema, lymphocyte infiltration and narrowing of the liver sinusoidal space, thickening of the alveolar wall septum and inflammatory cell infiltration in the lungs, swelling and congestion of renal tubular epithelial cells, renal interstitial cell hemorrhage, cellular edema and inflammatory cell infiltration in brain tissue compared with the normal control group mice. The tissue lesions of mice in the emodin administration group were alleviated compared to the virus control group. In mice in the emodin administration group, the cardiac muscle fibers were slightly broken and bleeding; hepatocyte edema and lymphocyte infiltration were slightly reduced; the alveolar wall septum was thickened in some lungs, but no bleeding was seen; the renal epithelium was locally infiltrated by inflammatory cells; epithelial cells were swollen and necrotic; and inflammatory cell infiltration and edema were slightly reduced in the brain tissue.

### 2.7. Changes in Serum Cytokines

The immune function status of the mice was assessed by measuring cytokines with ELISA kits, and the results are shown in Figure 7. Compared with normal control mice, PRV infection led to significant increases in TNF-α, IFN-γ, and IL-6 levels in the serum of mice (*p* < 0.05), while the level of IL-4 was significantly decreased. Compared with the virus control group, the levels of TNF-α, IFN-γ, and IL-4 in the serum of emodin-treated mice were significantly increased (*p* < 0.05), and the level of IL-4 was increased to the point of no significant difference from the normal control group. There was no significant difference in the serum level of IL-6 in the emodin administration group compared with the virus control group (*p* > 0.05).

## 3. Materials and Methods

### 3.1. Cells, Virus, and Reagents

Porcine kidney 15 (PK-15) cells were cultured in Dulbecco’s modified Eagle’s medium (DMEM, Gibco, Grand Island, NY, USA) supplemented with 1% penicillin/streptomycin and 10% fetal bovine serum (FBS, Gibco, NY, USA). The medium used for cytotoxicity and antiviral assays contained 2% FBS. The PRV TJ strain (GenBank accession: KJ789182.1) was kindly provided by Harbin Veterinary Research Institute, Chinese Academy of Agricultural Sciences. The virus was propagated in PK15 cells, and the 50% tissue culture infective dose (TCID50) was determined to be 10−7.04/mL. The virus was stored at −80 °C until use. Emodin at a purity of 98% was purchased from Dalian Meilun Biotechnology Co., Ltd. (Dalian, China) and dissolved in 1% dimethyl sulfoxide (DMSO) to prepare a stock solution.

### 3.2. Assays for Cytotoxicity and Inhibitory Activity

Cell viability was detected by Cell Counting Kit-8 (CCK-8, Bimake, Houston, TX, USA) to evaluate the cytotoxicity of emodin. Briefly, 1 × 10^5^ PK-15 cells/mL were seeded in 96-well tissue culture plates (Coster, Ashburn, VA, USA) and cultured at 37 °C in an atmosphere of 5% CO_2_ until the cells reached confluency. After removal of the growth medium, serial twofold dilutions of emodin ranging from 0.063 to 2 mg/mL were added to the plate, and the cell monolayer was incubated with different concentrations of emodin for 48 h. CCK-8 solution was added according to the instructions and incubated at 37 °C for 45 min. The OD values of each well at a 450 nm wavelength was measured using a microplate reader. The cell viability was expressed as a percentage of the cell viability of the control to evaluate the cytotoxicity of emodin. The 50% cytotoxic concentration (CC50) of emodin was calculated by GraphPad Prism 8.0 software. To determine the inhibitory activity of emodin on PRV, PK-15 cells were infected with 100 TCID50 PRV for 1 h. After washing twice with PBS, the cells were incubated with DMEM containing 2% FBS with or without emodin. After 48 h, the OD value of each well at a 450 nm wavelength was measured, and the virus inhibition rate was calculated to evaluate the inhibitory activity of emodin on PRV. Here, the PRV inhibition ratio = (OD450 value of emodin treated group–OD450 value of PRV control)/(OD450 value of cell control–OD450 value of PRV control) × 100%. The concentration for 50% of maximal effect (EC50) was calculated using GraphPad Prism 8.0 software. The selectivity index (SI) was evaluated by the following formula: SI = CC50/EC50.

### 3.3. Inhibitory Action Assay

Inactivation assay: Emodin was mixed with an equal volume of viral solution containing 100 TCID50 of PRV at 37 °C for 1 h. The mixture was added to PK-15 cells that had grown to 80–90% confluence and was incubated for 1 h at 37 °C. After washing, DMEM containing 2% FBS was added for incubation. The cells were collected after 40 h, and total DNA was extracted using a TIANamp Genomic DNA Kit (Tiangen Biotech, Beijing, China). The copy number of viral DNA in infected cells was analyzed by fluorescent quantitative PCR (FQ-PCR). The primer sequences were 5′-GCCGAGTACGACCTCTGCC-3′ (forward) and 5′-CGAGACGAACAGCAGCCG-3′ (reverse), and the probe sequence was 5′-HEX-CCGCGTGCACCACGAAGCCT-BHQ1-3′. A standard curve was initially generated using a purified plasmid containing the gI gene. The FQ-PCR conditions were 94 °C for 5 min, 94 °C for 35 s, and 60 °C for 35 s (40 cycles).

Cell pretreatment assay: Emodin was added to PK-15 cells and incubated at 37 °C for 4 h. The medium was then removed, and the cells were infected with 100 TCID50 of PRV for 1 h. After washing twice with PBS, DMEM containing 2% FBS was added to the cells for 40 h. Total DNA was extracted and subjected to a FQ-PCR assay.

Virus adsorption assay: PK-15 cells were precooled at 4 °C for 1 h, and 100 TCID50 PRV was added to the cells in the presence of emodin at 4 °C for 1 h. The cells were washed with ice-cold PBS, and DMEM containing 2% FBS was added for 40 h. Total DNA was extracted and subjected to a FQ-PCR assay.

Virus entry assay: PK-15 cells were precooled at 4 °C for 1 h and then infected with 100 TCID50 of PRV at 4 °C for 1 h. After washing, emodin was added to the cells for incubation for 1 h at 37 °C. The cells were then washed with PBS, and DMEM containing 2% FBS was added at 37 °C for 40 h. Total DNA was extracted and subjected to a FQ-PCR assay.

Viral replication assay: PK15 cells were challenged with 100 TCID50 PRV for 1 h at 37 °C. After the removal of unbound viruses, the cells were incubated with DMEM containing emodin at 37 °C for 40 h. Total DNA was extracted and subjected to a FQ-PCR assay.

### 3.4. Gene Expression Assay

The cells were treated as described in the inhibitory activity assay. Infected cell samples were harvested at 3, 6, 12, 24, and 48 hpi. The total RNA of cells was extracted using the RNA Easy Fast Tissue/Cell Kit (TIANGEN, Beijing, China), and cDNA was obtained by reverse transcription using FastKing-RT SuperMix (TIANGEN, China) for qPCR to detect the relative expression levels of genes. qPCR was performed on an ABI 7500 instrument using SYBR Green qPCR Master Mix (Bimake, TX, USA). The primers used for Qpcr are shown in Table 1. The reaction thermal conditions were in predenaturation at 95 °C for 5 min, then a three-step cycle procedure (95 °C for 15 s, 55 °C for 30 s and 72 °C for 30 s) for 40 cycles, then 95 °C for 15 s, 60 °C for 60 s, 95 °C for 15 s. Each reaction was performed in triplicate, the housekeeping gene β-actin was used to normalize for differences in total Cdna levels in samples, and the 2^−ΔΔCT^ method was used to calculate the PRV gene Mrna expression level of samples.

### 3.5. Western Blotting

PK15 cells were treated with emodin or control vehicle after PRV infection. After 36 h of incubation, cell samples were harvested for total cellular protein extraction and lysed with protein extraction buffer. The proteins were separated by 10% SDS–PAGE and transferred to a nitrocellulose membrane. The membrane was blocked with 5% fat-free milk for 2 h at room temperature (RT) and then incubated with antibodies. Mouse monoclonal anti PRV-Gb, mouse monoclonal anti PRV-Gd, and mouse monoclonal anti β-actin antibodies were used as the primary antibodies, and HRP-conjugated goat anti-mouse IgG was used as the secondary antibody. The protein bands on the membranes were then observed using an ECL™ detection system (Bio-Rad, Hercules, CA, USA), and the signals were analyzed using ImageJ software (National Institutes of Health, Bethesda, MD, USA). The expression of β-actin protein was used to normalize the difference between gB and gD protein levels in samples.

### 3.6. Molecular Docking

AUTODOCK 4.2.6 was used for molecular docking analysis of emodin and gB and gD proteins. AUTOGRID supporting software was used to calculate the atomic affinity potential energy. PyMOL was used to visualize the results. The chemical structure of emodin (MOLID: 000472) was downloaded from the PubChem database (https://pubchem.ncbi.nlm.nih.gov/ accessed on 10 July 2023). The X-ray structures of gB (PDB ID: 5YS6) and gD (PDB ID: 5 × 5 V) were downloaded from the PDB protein structure database (https://www.pdbus.org/ accessed on 8 December 2022).

### 3.7. Cell Apoptosis Assay

Cell apoptosis was analyzed with an Annexin V-FITC apoptosis detection kit (Meilun Biotechnology Co., Ltd., Dalian, China) by flow cytometry. Briefly, PK15 cells infected with PRV were treated with emodin for 36 h. The cell culture medium was collected, added to digested adherent cells, and centrifuged at 1000× *g* for 5 min. After the cell pellet was washed with PBS and resuspended by adding 500 μL binding solution, 5 μL annexin V-FITC and 5 μL PI were added for 10 min in the dark at RT. Cell apoptosis was measured by flow cytometry.

### 3.8. Mice Viral Challenge Protection Test

Thirty female Kunming mice at weights of 20 ± 2 g were purchased from the Changsheng Experimental Animal Center. After adaptive feeding for 1 week, the mice were randomly divided into three groups: the normal control group, virus control group, and emodin administration group. Mice in the emodin administration group were intraperitoneally injected with emodin at a dose of 50 mg/kg once a day for 3 days, while mice in the normal control group and virus control group were injected with the same volume of DMEM solution. On the 4th day, mice except those in the normal control group were infected by intramuscular injection of 0.1 mL DMEM containing 10,000 TCID50 PRV. After 1 h of challenge, the mice in the emodin administration group were intraperitoneally injected with emodin at a dose of 50 mg/kg once a day for four days. Simultaneously, the normal control group and virus control group were injected with DMEM. The weight, behavior and general health condition of the mice were recorded daily. On the third day after challenge, three mice from each group were randomly selected for blood collection and dissection; the visceral lesions of mice were observed; and the heart, liver, lung, kidney, brain tissue and other organs were collected. The remaining mice were monitored simultaneously to determine the survival rate during the remaining experimental period. Some of the collected mouse organ tissues were homogenized in normal saline to extract DNA to determine the viral load in the tissues. The other portions were fixed in 4% paraformaldehyde and then processed for tissue sectioning and histopathological examination. The collected blood samples obtained by ocular blood collection were coagulated naturally for 20 min at RT and centrifuged at 3000 rpm for 15 min. The supernatant was carefully collected for detection of cytokines (IL-4, IL-6, TNF-α and IFN-γ) in the serum using ELISA kits (Meilun Biotechnology Co., Ltd., Dalian, China).

### 3.9. Statistical Analysis

All cell experiments were repeated at least three times, and the results are presented as the mean ± standard deviation (SD). The statistical significance of the data was assessed by two-tailed Student’s *t* test with GraphPad Prism 5.0 software (La Jolla, CA, USA). A *p* value of <0.05 was considered statistically significant.

## 4. Discussion

PRV infection can lead to reproductive disorders in pregnant sows; fatal encephalitis, neurological disorders and high mortality in newborn piglets; and respiratory distress and growth disorders in piglets [24]. Once PRV has infected a pig population, it is difficult to eliminate, seriously affecting the healthy development of the pig breeding industry. Vaccine immunization is the main strategy to prevent PRV infection, and the Bartha-K61 vaccine has played an important role in preventing and controlling PR [25]. However, since 2011, the emergence of new PRV variant strains has greatly increased the PRV infection rate in China’s pig farms, causing major economic losses in the pig industry. The variant strains are highly pathogenic in pigs of different ages and induce different degrees of clinical symptoms, which means that the traditional Bartha-K61 vaccine can only provide limited immune protection [26,27,28]. In addition, reports of human infection with PRV have also attracted global attention [29]. Therefore, it is particularly important to find more PRV prevention and control strategies, and developing effective anti-PRV drugs is a novel means to prevent and control PRV infection. The efficacy of therapy depends on the binding of a drug to its cognate target, and optimizing the target binding of drugs in cells is often challenging. However, emodin has potential off-target effects, which should be addressed in future studies [30]. Based on the extensively reported antiviral properties of emodin, the antiviral activities of emodin against PRV were investigated in this study. The viability of PK-15 cells in the presence of emodin was detected by CCK-8, and the CC50 of emodin was calculated to be 0.7009 mg/mL. Then, we further studied the anti-PRV activity of emodin within a safe concentration range, and the results showed that emodin significantly inhibited the proliferation of PRV in PK-15 cells in a concentration-dependent manner, with good PRV inhibitory activity. Wang [31] et al. reported that piceatannol had an anti-PRV effect, its SI value was 3.68, and the survival rate of mice was 14.3%. The SI value of emodin was 5.52, and the survival rate of mice was 28.5%. In contrast, emodin has good anti-PRV activity.

To further study the antiviral mechanism of emodin, assays were performed at different stages of viral infection. The results showed that emodin could significantly reduce the virus copy number of infected cells and inhibit virus proliferation in the virus adsorption stage, cell entry stage, and intracellular replication stage; but, it could not directly inactivate virions. During the PRV life cycle, virions are first attached to cells through the interaction of gC with heparan sulfate proteoglycans in the extracellular matrix [32]. Next, PRV gD binds to specific cellular receptors to stabilize the interaction between virions and cells. Simultaneously, PRV gB, gH, and gL mediate the fusion of the viral envelope and cytoplasmic membrane [33,34], allowing the PRV viral capsid and envelope to penetrate host cells. When PRV DNA enters the host nucleus, it can regulate cellular RNA polymerase II to initiate viral gene transcription. Gene transcription is a key step in the viral replication phase, and PRV infection is mainly achieved by controlling the transcription level [35]. Kaempferol can prevent PRV infection by inhibiting the transcription levels of the IE180, EP0, TK and LAT genes of PRV, and luteolin can inhibit viral DNA replication by affecting the transcription levels of IE180, EP0 and other replication-related genes of PRV [35]. Our results showed that emodin could also inhibit viral DNA replication and delay PRV infection by inhibiting the transcription levels of the PRV immediate early gene IE180, early gene EP0 and other PRV infection-related genes (US6, UL44, UL29, and UL27). In addition, viral proteins expressed across the viral life cycle can be potential targets of antiviral drugs. PRV gD and gB glycoproteins are encoded by the US6 and UL27 genes, respectively, and are indispensable key proteins in the fusion process of virions and host cell membranes [23,36]. Quercetin can interact with the gD protein, which plays an important role in the virucidal properties of quercetin and inhibition of PRV infection [13]. Luteolin can inhibit PRV proliferation and cell–cell transmission by inhibiting the expression of the gB gene and protein [14]. This study found that emodin can also inhibit the expression of the gD and gB genes, block the synthesis of the PRV gB protein and gD protein, and inhibit virus proliferation. In addition, through molecular docking to predict the binding site of the drug and virus, it was found that emodin could form stable hydrogen bond structures with the PRV gB protein and gD protein. It was speculated that the interaction between the drug and the protein changed the spatial structure of the gB protein and gD protein, thus inhibiting the proliferation of PRV virions.

Natural products have multitarget and multichannel antiviral activities. In addition to directly affecting the virus replication cycle and inhibiting virus proliferation, they can also play antiviral roles in other ways [37]. As a cellular defense mechanism, apoptosis plays an important role in preventing the spread of virus in the early stages of virus infection, while it enhances the replication and export of virus in the later stages of virus infection [38]. Virus infection usually leads to cell apoptosis; thus, many viruses inhibit the premature death of infected cells through anti-apoptosis mechanisms in the early stages of infection and promote cell apoptosis in the later stages to safeguard the process of virus proliferation and increase the production of offspring viruses [39,40,41]. PRV also uses this mechanism for cell apoptosis in the early stages of alphaherpesvirus infection, and the protein kinase encoded by US3 plays an important role in protecting early-infection cells from apoptosis and in promoting cell survival [42,43]. However, the inhibitory effect of herpesvirus on cell apoptosis is only temporary. Studies have found that PRV can activate the expression of Caspase-3 in the later stages of infection and mediate cell apoptosis in vitro and in vivo to ensure the generation of offspring virus [44]. Curcumin, myricetin and luteolin can effectively inhibit virus infection by blocking PRV-induced apoptosis [45,46]. Liu et al. found that emodin can reduce CVB4-induced apoptosis in vitro and in vivo [47]. Therefore, inhibition of virus-induced apoptosis may be a potential target for anti-PRV drugs. In this study, the effect of emodin on the apoptosis of host cells in the later stages of PRV infection was detected by flow cytometry. The results confirmed that PRV infection could induce the apoptosis of host cells. Moreover, emodin inhibited the PRV-induced apoptosis by playing an anti-apoptotic role, thereby limiting the production of virus particles in the offspring and inhibiting virus proliferation.

The PRV-inhibitory activity of emodin was evaluated by administration before and after viral challenge in mice. The study found that emodin has a beneficial protective effect on PRV-infected mice. The specific results were as follows: compared to the untreated virus control group, emodin administration delayed the onset time of clinical symptoms of infected mice, prolonged the average survival time, and improved the survival rate of infected mice. It is speculated that the use of drugs may slow the virus proliferation rate in mice. Viral load is a direct parameter to evaluate the antiviral effect in vivo, which can reflect the replication of virus in different organs [48]. This study found that the viral load in the brains of mice in the virus control group was the highest, which may be related to the neurophilic properties of alphaherpesviruses [49]. Emodin administration significantly reduced the viral load in the brain, liver and lung tissues of infected mice; emodin has anti-inflammatory, diuretic and hepatoprotective and neuroprotective effects. It is speculated that emodin can inhibit the replication of PRV in mouse organs. Moreover, histopathological observation confirmed that emodin had a good antiviral effect on PRV-infected mice.

The host inflammatory response is the first line of defense to prevent the spread of viral infection, and cytokines are the regulatory factors of the host inflammatory response. TNF-α, IFN-γ, and IL-6, as important proinflammatory cytokines, can help the body fight viral infection and tissue damage by regulating the inflammatory response [50,51,52,53]. Therefore, the level of these cytokines is crucial to combat viral infection. In this study, the levels of the cytokines IL-6, TNF-α and IFN-γ in the serum of mice increased significantly after viral challenge, indicating that PRV infection activated the innate immunity of mice. Emodin administration significantly increased TNF-α and IFN-γ levels in the serum of infected mice. It is presumed that emodin can enhance the inflammatory response and inhibit virus replication by increasing the levels of TNF-α and IFN-γ in the early stages of virus infection. As an anti-inflammatory cytokine, IL-4 is mainly produced by Th2 cells. In host immunity, it not only regulates the immune function of macrophages but also plays an important role in promoting the development of Th2 cells and inhibiting the growth of Th1 cells [54,55]. In this study, the level of IL-4 in the serum of infected mice decreased, and emodin administration significantly rescued its level. It is speculated that emodin can regulate the Th1/Th2 balance by regulating the level of IL-4 to protect against PRV infection.

## 5. Conclusions

In conclusion, emodin showed good PRV inhibitory activity in vitro and in vivo. Emodin can inhibit virus proliferation in the intracellular replication stages and can exert antiviral effects in vitro by inhibiting PRV gene expression and PRV gB and gD protein expression, as well as alleviating PRV-induced apoptosis. In addition, emodin can protect PRV-infected mice from lethal effects by regulating the levels of cytokines in the serum of infected mice. These findings suggest that emodin may be a potential antiviral candidate for controlling PRV infection in the future.

## Figures and Tables

**Figure 1 molecules-28-06567-f001:**
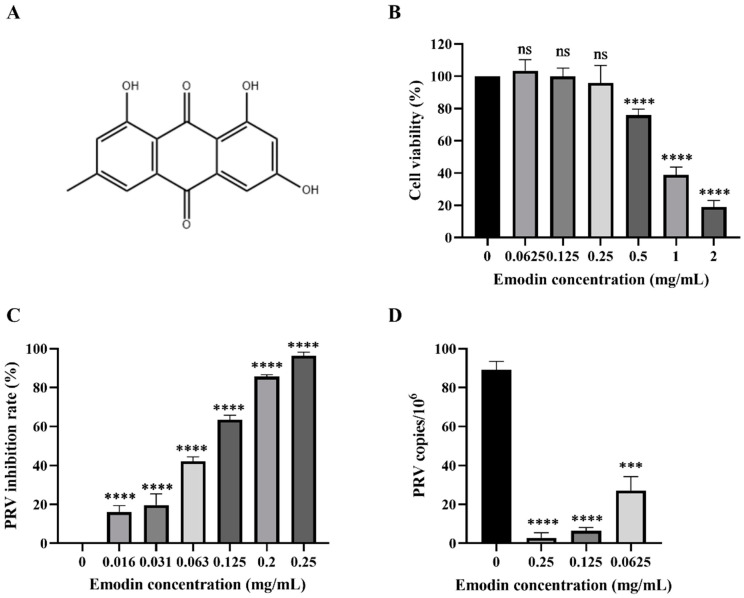
Emodin inhibits PRV proliferation in PK-15 cells. (**A**) Emodin chemical structure. (**B**) Emodin cytotoxicity; the viability of PK-15 cells was determined by CCK-8 method at concentrations of 0.0625–2.0 mg/mL emodin, and the cell survival rate was expressed as the percentage of cell viability in the control group. (**C**) The inhibition rate of emodin on PRV was detected by CCK-8 method to calculate the inhibition rate of emodin on PRV. The inhibition rate (%) = (OD450 value of emodin treated group–OD450 value of PRV control)/(OD450 value of cell control–OD450 value of PRV control) ×100%. (**D**) FQ-PCR was used to detect the viral DNA copy number of PRV infected cells by emodin. Compared with the control group, ns: *p* > 0.05, *** *p* < 0.001, **** *p* < 0.0001.

**Figure 2 molecules-28-06567-f002:**
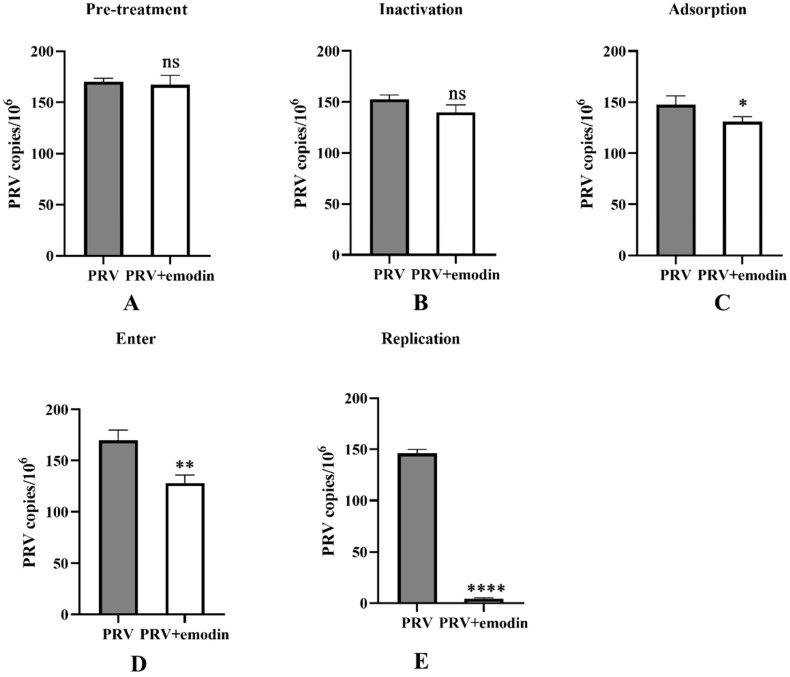
Effect of emodin on the stages of the PRV life cycle. (**A**) The pretreatment effect of emodin against PRV. (**B**) The inactivation effect of emodin against PRV. (**C**) The inhibitory effect of emodin on PRV adsorption. (**D**) The inhibition of PRV entry by emodin. (**E**) The inhibitory effect of emodin on PRV replication; the concentration of emodin was 0.25 mg/mL. Compared with PRV, ns: *p* > 0.05, * *p* < 0.05, ** *p* < 0.01, and **** *p* < 0.0001.

**Figure 3 molecules-28-06567-f003:**
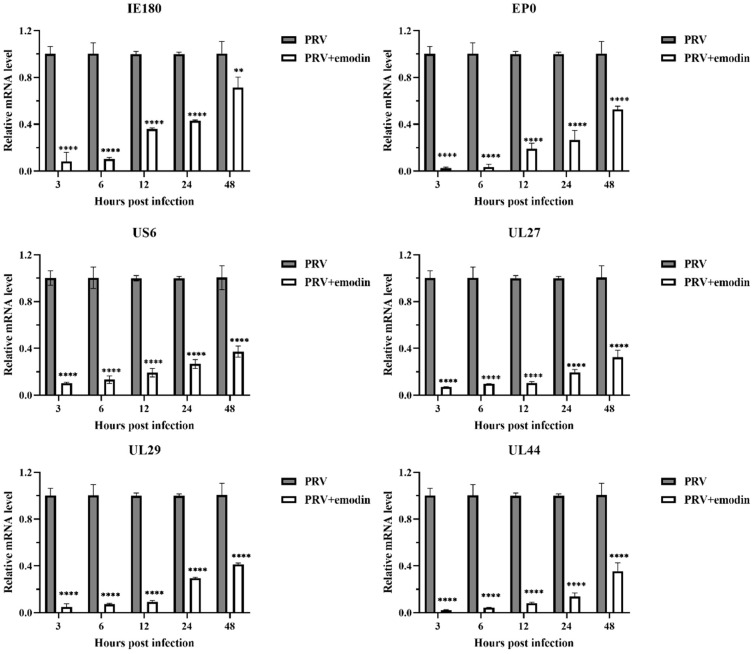
Emodin inhibits PRV gene expression. The gene expression levels of PRV at 3, 6, 12, 24, and 48 hpi in the presence or absence of emodin (0.25 mg/mL). Compared with PRV, ** *p* < 0.01, **** *p* < 0.0001.

**Figure 4 molecules-28-06567-f004:**
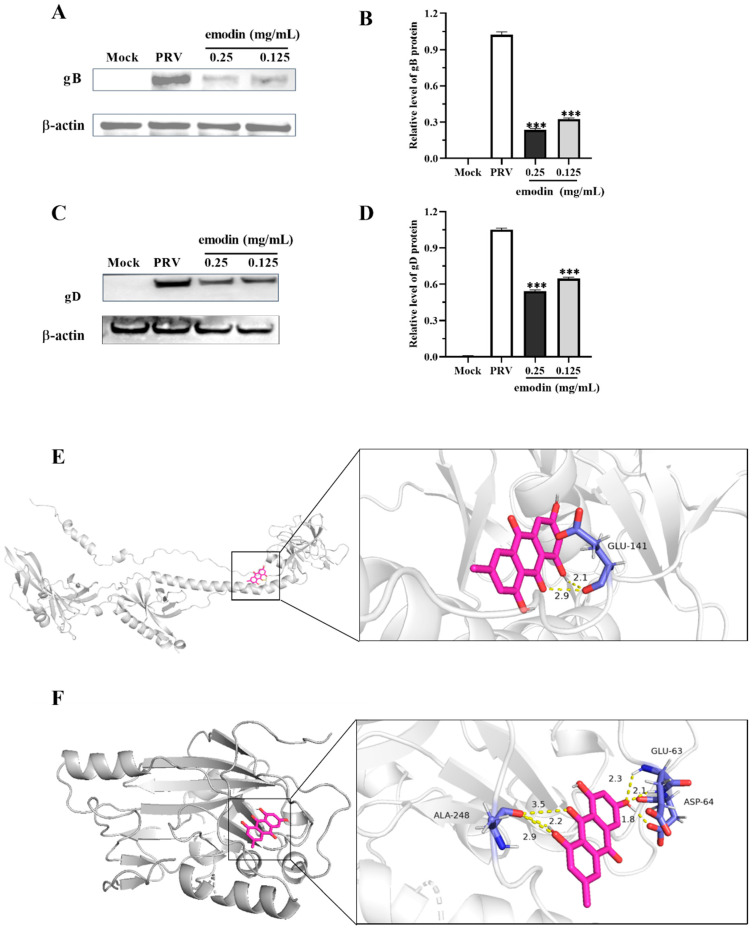
Effect of emodin on PRV gB and gD protein expression. PK15 cells were treated with or without emodin after PRV infection. After 36 h of culture, the cells were collected, and protein extraction was performed for Western blotting to calculate the relative expression of gB protein (**A**) and gD protein (**C**). (**B**) Quantification of the gray value of the gB protein. (**D**) Quantification of the gray value of the gD protein. (**E**) Optimal binding results of the PRV gB protein to emodin. (**F**) Optimal binding results of the PRV gD protein to emodin. The magenta rod structure represents emodin, and the PRV gB protein and gD protein are represented by cartoons. Compared with PRV, *** *p* < 0.001.

**Figure 5 molecules-28-06567-f005:**
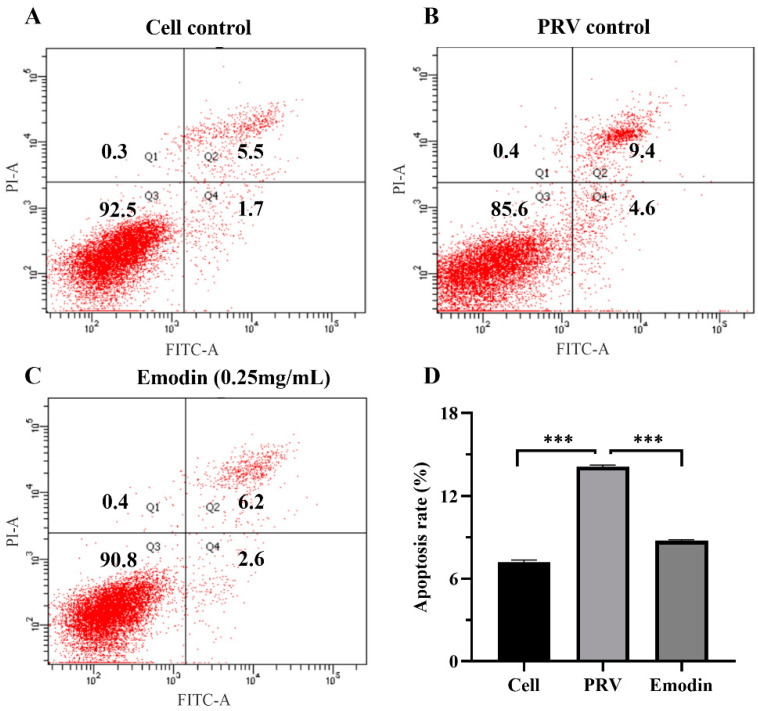
Emodin alleviated apoptosis induced by PRV infection. PK-15 cells were infected with PRV and treated with emodin for 36 h, stained with Annexin-V and PI, and analyzed using flow cytometry. (**A**) Apoptosis rate in the cell control group. (**B**) Apoptosis rate of the PRV control group. (**C**) Apoptosis rate of the emodin-treated group. (**D**) Column bar graph of apoptosis. (*** *p* < 0.001).

**Figure 6 molecules-28-06567-f006:**
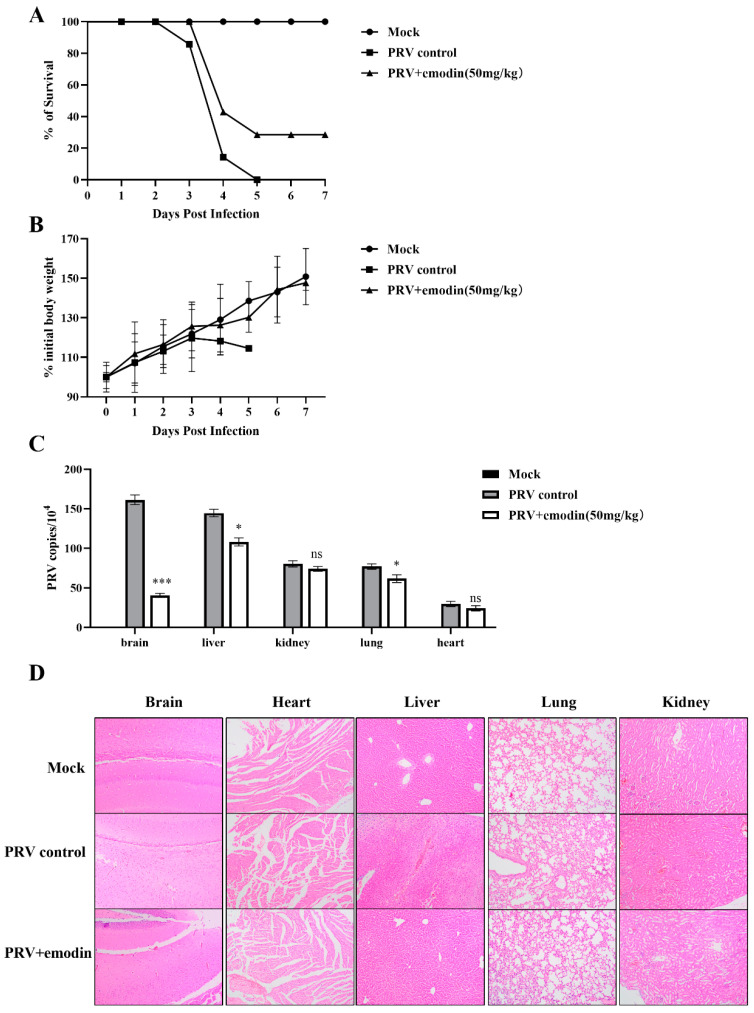
Emodin inhibits PRV proliferation in mice. (**A**) Survival rate of mice in each group. The survival rate of each group was calculated according to the following formula: survival rate = number of surviving mice/total number of mice. (**B**) Body weight changes of mice in each group. (**C**) Viral load in tissues and organs of mice in each group. On the third day after PRV challenge, the heart, liver, lung, kidney and brain tissues of mice were collected, and viral DNA was extracted. The viral copy number in each organ tissue was detected by FQ-PCR. Compared with the PRV control group, ns: *p* > 0.05, * *p* < 0.05, *** *p* < 0.001. (**D**) HE stained pathological sections of mouse brain, heart, liver, lung, and kidney tissues.

**Figure 7 molecules-28-06567-f007:**
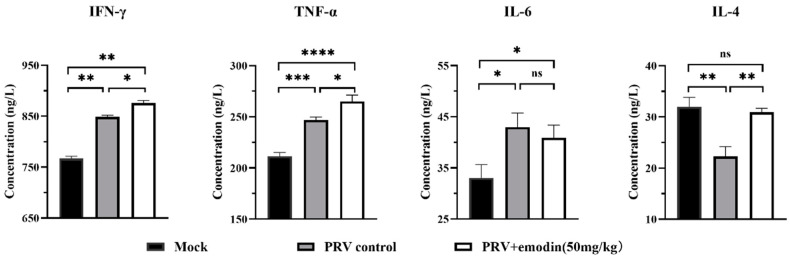
The cytokine concentration in serum of mice in each group. On the 3rd day after PRV infection, blood was collected from three mice randomly selected from each group and used to determine levels of four cytokines in the serum of mice. (ns: *p* > 0.05, * *p* < 0.05, ** *p* < 0.01, *** *p* < 0.001, **** *p* < 0.001).

**Table 1 molecules-28-06567-t001:** Primer sequences for real-time PCR.

Gene Name	Sequence (5′-3′)
UL44-F	CGTCAGGAATCGCATCA
UL44-R	CGCGTCACGTTCACCAC
IE180-F	CGCTCCACCAACAACC
IE180-R	TCGTCCTCGTCCCAGA
UL29-F	AGAAGCCGCACGCCATCACC
UL29-R	GGGAACCCGCAGACGGACAA
EP0-F	GGGCGTGGGTGTTT
EP0-R	GCTTTATGGGCAGGT
US6-F	AACATCCTCACCGACTTCA
US6-R	CGTCAGGAATCGCATCA
UL27-F	TCGTCCACGTCGTCCTCTTCG
UL27-R	CGGCATCGCCAACTTCTTCC
β-actin-F	TGCGGGACATCAAGGAGAA
β-actin-R	AGGAAGGAGGGCTGGAAGA

## Data Availability

The original contributions presented in the study are included in the article, and further inquiries can be directed to the corresponding author(s).

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
