# Peer review of "Emodin as an Inhibitor of PRV Infection In Vitro and In Vivo"

_molecules, 2023, doi:10.3390/molecules28186567_

Round 1
Reviewer 1 Report
Comments and Suggestions for Authors
1. The study mentions the use of PK15 cells and mice for the experiments. However, this study does not provide information on the number of samples or animals used in each group or experiment. Why authors have chosen only one kidney cell line for their experiment?
2. While the work mentions that emodin inhibits PRV gene expression and viral replication, it does not provide a detailed explanation of the underlying molecular mechanisms. The introduction section should include information about the relevance of choosing PRV gB and gD proteins as targets for studying the antiviral activity of emodin.
3. Present work does not mention any comparison of emodin with other known antiviral agents against PRV. Without such comparisons, it is difficult to assess the relative efficacy and potential advantages of emodin as a treatment option.
4. Although the study briefly mentions the cytotoxicity of emodin, it does not provide detailed information on its safety profile or potential side effects.
5. The work primarily focuses on in vitro and in vivo experiments using cell lines and animal models. However, there is no mention of clinical trials or human studies to validate the effectiveness of emodin as an antiviral agent against PRV.
6. The present study mentions the survival rate and body weight changes of mice treated with emodin. However, there is no information on the long-term effects or durability of the observed antiviral effects.
7. Have you considered using other docking algorithms or performing cross-validation with different algorithms to validate the docking results?
8. How proteins and ligands were prepared for docking study? Authors have not properly justified their computational study.
9. Why not authors have considered the potential off-target effects of emodin? It would be valuable to discuss any potential interactions with other proteins or unintended effects that may arise from emodin treatment.
10. What is the broader applicability of emodin as an antiviral agent?
11. The study states that blood samples were obtained by "ocular blood collection." However, it is more common and appropriate to collect blood samples from mice through methods such as retro-orbital bleeding or tail vein puncture. Why authors have not performed those?
12. The study does not provide information on the number of mice used in each group.
13. Why authors have not mention the appropriate positive and negative controls were included in the ELISA assay? The study have not mention whether the ELISA assays were performed in duplicate or triplicate.
14. The study mentions that the cell apoptosis was measured by flow cytometry after a 10-minute incubation with Annexin V-FITC and PI. However, the standard protocol for Annexin V-FITC apoptosis detection usually involves a longer incubation period (15-30 minutes) to allow for proper staining and detection of apoptotic cells. Authors need to justify this.
15. What specific flow cytometer is used for the apoptosis analysis?
16. What specific assay was used to measure cell viability?
17. The present study states that the CC50 (50% cytotoxic concentration) of emodin was calculated by GraphPad Prism 8.0 software. However, the CC50 is typically determined experimentally by performing a dose-response curve and calculating the concentration of emodin that causes a 50% reduction in cell viability. It is not clear if the CC50 value mentioned was determined experimentally or calculated based on the EC50 value.
18. Why authors have not performed any statistical analysis on the qPCR data?
Comments on the Quality of English Language
1. The manuscript should be formatted properly, with commas, full stops, brackets, spellings, and citations all in their respective places.
2. The manuscript needs extensive revision for language and grammar.
Author Response
Response to Reviewer 1Comments
Dear Editors and Reviewer,
Thank you for your work and comment. The comments and suggestions are very helpful for revising and improving our paper and research. We have studied every comment carefully and made corrections one by one. For your valuable comments, words in red are the revisions we have made in the manuscript. Generally, this study made the following responses:
Point 1: The study mentions the use of PK15 cells and mice for the experiments. However, this study does not provide information on the number of samples or animals used in each group or experiment. Why authors have chosen only one kidney cell line for their experiment?
Response 1: Thank you very much for your comments. A total of 30 mice were used for the in vivo experiments. There were 10 mice in each group. PRV was able to induce PK-15 to produce CPE in vitro. Therefore, PRV was used as a model to study the anti-PRV effect of picrotaxol.
Point 2: While the work mentions that emodin inhibits PRV gene expression and viral replication, it does not provide a detailed explanation of the underlying molecular mechanisms. The introduction section should include information about the relevance of choosing PRV gB and gD proteins as targets for studying the antiviral activity of emodin.
Response 2: Thank you very much for your questions. We have only preliminarily explored the effects of emodin on PRV gene expression and viral replication. We will continue to investigate the underlying molecular mechanisms in the future.
Point 3: Present work does not mention any comparison of emodin with other known antiviral agents against PRV. Without such comparisons, it is difficult to assess the relative efficacy and potential advantages of emodin as a treatment option.
Response 3: Thank you very much for your valuable suggestions. I have added a comparison to known anti-PRV drugs in the discussion section in line 384-387.
Point 4: Although the study briefly mentions the cytotoxicity of emodin, it does not provide detailed information on its safety profile or potential side effects.
Response 4: Thank you very much for your comments. I have already added that to the preface in line 77-78.
Point 5: The work primarily focuses on in vitro and in vivo experiments using cell lines and animal models. However, there is no mention of clinical trials or human studies to validate the effectiveness of emodin as an antiviral agent against PRV.
Response 5: Thank you very much for your questions. This study mainly explored the inhibitory effect of emodin on PRV, and provided some possibilities for the development of anti-PRV drugs in the future. So there are no human trials involved.
Point 6: The present study mentions the survival rate and body weight changes of mice treated with emodin. However, there is no information on the long-term effects or durability of the observed antiviral effects.
Response 6: We appreciate this commet very much. In this experiment, the survival rate of mice in emodin administration group was 28.5%, and that in PRV challenge group was 0%. In comparison, emodin has a certain effect. The weight change curve of mice showed that compared to the normal control group, the weight of mice in the virus control group continuously decreased after symptoms appeared on the third day. The average body weight loss of mice in the emodin administration group slowed on the 4th and 5th days and showed a significant upward trend after the 6th day.
Point 7: Have you considered using other docking algorithms or performing cross-validation with different algorithms to validate the docking results?
Response 7: Thank you very much for your valuable suggestions. In the future research, we will study and apply in this aspect.
Point 8: How proteins and ligands were prepared for docking study? Authors have not properly justified their computational study.
Response 8: Thank you very much for your questions. The proteins and ligands used in the molecular docking were downloaded from the website.
Point 9: Why not authors have considered the potential off-target effects of emodin? It would be valuable to discuss any potential interactions with other proteins or unintended effects that may arise from emodin treatment.
Response 9: Thank you very much for your valuable suggestions. I've added that to the discussion section in line 381-383.
Point 10: What is the broader applicability of emodin as an antiviral agent?
Response 10: Thank you very much for your questions. Emodin has a wide range of pharmacological properties, including antiviral, antibacterial, anti-inflammatory, antitumor, immunosuppressive, and neuroprotective properties[1]. Especially in terms of antiviral properties, it has been shown to have a variety of antiviral activities, including against herpes simplex virus, SARS coronavirus, influenza A virus and porcine reproductive and respiratory syndrome virus[2-5].
- Li Q, Gao J, Pang X, Chen A, Wang Y. Molecular mechanisms of action of emodin: as an anti-cardiovascular disease drug. Front Pharmacol. (2020) 11: 559607. doi: 10.3389/fphar.2020.559607
- Ho TY, Wu SL, Chen JC, Li CC, Hsiang CY. Emodin blocks the SARS coronavirus spike protein and angiotensin-converting enzyme 2 interaction. Antiviral Res. (2007) 74: 92-101. doi: 10.1016/j.antiviral.2006.04.014
- Hsiang CY, Ho TY. Emodin is a novel alkaline nuclease inhibitor that suppresses herpes simplex virus type 1 yields in cell cultures. Br J Pharmacol. (2008) 155: 227-35. doi: 10.1038/bjp.2008.242
- Dai JP, Wang QW, Su Y, Gu LM, Zhao Y, Chen XX, et al. Emodin inhibition of influenza a virus replication and influenza viral pneumonia via the nrf2, tlr4, p38/jnk and nf-kappab pathways. Molecules. (2017) 22. doi: 10.3390/molecules22101754
- Xu Z, Huang M, Xia Y, Peng P, Zhang Y, Zheng S, et al. Emodin from aloe inhibits porcine reproductive and respiratory syndrome virus via toll-like receptor 3 activation. Viruses. (2021) 13. doi: 10.3390/v13071243
Point 11: The study states that blood samples were obtained by "ocular blood collection." However, it is more common and appropriate to collect blood samples from mice through methods such as retro-orbital bleeding or tail vein puncture. Why authors have not performed those?
Response 11: Thank you very much for your questions. Mouse eyeball blood collection experiments are usually conducted to obtain whole blood samples of mice for hematology, immunology, or blood biochemistry experiments. Compared with other blood collection methods, eyeball blood collection has the advantages of simple operation, rapid operation, no need for a large number of animals, and is not limited by weight and body size.
Point 12: The study does not provide information on the number of mice used in each group.
Response 12: Thank you very much for your questions. A total of 30 mice were used in this experiment, with 10 mice in each group.
Point 13: Why authors have not mention the appropriate positive and negative controls were included in the ELISA assay? The study have not mention whether the ELISA assays were performed in duplicate or triplicate.
Response 13: Thank you very much for your questions. Because no specific anti-PRV drug has been officially released, this trial did not have a positive control. The ELISA test in the study was run in triplicate.
Point 14: The study mentions that the cell apoptosis was measured by flow cytometry after a 10-minute incubation with Annexin V-FITC and PI. However, the standard protocol for Annexin V-FITC apoptosis detection usually involves a longer incubation period (15-30 minutes) to allow for proper staining and detection of apoptotic cells. Authors need to justify this.
Response 14: Thank you very much for your questions. We performed the procedure according to the instructions of Annexin V-FITC apoptosis kit (Dalian Bolin Biotechnology Co., LTD., Dalian, China).
Point 15: What specific flow cytometer is used for the apoptosis analysis?
Response 15: We appreciate this commet very much. Cell apoptosis was analyzed using a sorting flow cytometer (FACSAria Ⅲ). The manufacturer is BD Company of the United States.
Point 16: What specific assay was used to measure cell viability?
Response 16: We appreciate this commet very much. We used a CCK8 kit and a microplate reader to measure cell viability.
Point 17: The present study states that the CC50 (50% cytotoxic concentration) of emodin was calculated by GraphPad Prism 8.0 software. However, the CC50 is typically determined experimentally by performing a dose-response curve and calculating the concentration of emodin that causes a 50% reduction in cell viability. It is not clear if the CC50 value mentioned was determined experimentally or calculated based on the EC50 value.
Response 17: We appreciate this commet very much. The CC50 and EC50 were calculated by GraphPad Prism 8.0 software.
Point 18: Why authors have not performed any statistical analysis on the qPCR data? Response 18: We appreciate this commet very much. The qPCR data were analyzed by t-test using GraphPad Prism 5.0 software.
Point 19: The manuscript should be formatted properly, with commas, full stops, brackets, spellings, and citations all in their respective places.
Response 19: Thank you very much for your valuable suggestions. I apologize for the minor errors in the manuscript. I have made the appropriate changes.
Point 20:The manuscript needs extensive revision for language and grammar.
Response 20: Thank you very much for your valuable suggestions. This manuscript was polished by a professional organization before resubmission.
We try our best to improve the manuscript and have made some changes marked in red in revised manuscript which will not influence the content and framework. We appreciate Editors and Reviewer’ warm work earnestly, and hope the revision will meet with approval. Once again, thank you very much for your comments and suggestions.
Kind regards,
Yan Zhu
Corresponding author.

Reviewer 2 Report
Comments and Suggestions for Authors
Present manuscript deals with the biological evaluation of very-well studied anthraquinone family member Emodin as an inhibitor of PRV infection. Authors successfully showed that Emodin inhibits the proliferation of PRV in PK15 cells with low mg/ml concentration in vitro enzymatic assay. Moreover, this molecule was tested at different stages of viral infection.
1. Figure 2: Each graph should be designated (A, B,….E) according to the running title of the figure.
2. Section 3.1: Authors showed that Emodin significantly inhibits the PRV replication. Although, it shows dose-dependent decrease in the number of virus copy (Figure 1C), it will be helpful to have positive and negative control for this study.
3. Section 3.2: Figure 2C, author claims that emodin significantly reduced the viral DNA copy numbers and inhibits the PVR at “absorption” stage. This has to be modified in the running text with moderate reduction. In fact, except replication stage there is no significant inhibition by the Emodin.
4. For Figure 3: What is the reliability of the experiments. Authors should provide details in running text or figure legend.
5. Section 3.4: Figure 4A-B: missing in running text.
6. Figure 7: Author claims that there is a significant increase in TNF-alpha, IFN-gamma, and IL-6 levels. This sentence has to be redrafted correctly as there is moderate effect on these cytokine concentrations.
Comments on the Quality of English Language
Minor redrafting is required for English language.
Author Response
Response to Reviewer 2Comments
Dear Editors and Reviewer,
Thank you for your work and comment. The comments and suggestions are very helpful for revising and improving our paper and research. We have studied every comment carefully and made corrections one by one. For your valuable comments, words in red are the revisions we have made in the manuscript. Generally, this study made the following responses:
Point 1: Figure 2: Each graph should be designated (A, B,….E) according to the running title of the figure.
Response 1: We appreciate this commet very much. I'm sorry for this small mistake. I have modified Figure 2.
Point 2: Section 3.1: Authors showed that Emodin significantly inhibits the PRV replication. Although, it shows dose-dependent decrease in the number of virus copy (Figure 1C), it will be helpful to have positive and negative control for this study.
Response 2: We appreciate this commet very much. This assay procedure consisted of cell and virus control groups and was performed in triplicate for each group.
Point 3: Section 3.2: Figure 2C, author claims that emodin significantly reduced the viral DNA copy numbers and inhibits the PVR at “absorption” stage. This has to be modified in the running text with moderate reduction. In fact, except replication stage there is no significant inhibition by the Emodin.
Response 3: Thank you very much for your valuable suggestions. I have made the corresponding changes in line 19-20, 245-246 and 469.
Point 4: For Figure 3: What is the reliability of the experiments. Authors should provide details in running text or figure legend.
Response 4: Thank you very much for your valuable suggestions. This part of the test is mainly designed according to the following references.
Cai, X.; Shao, Y.; Wang, Z.; Xu, Y.; Ren, Z.; Fu, L.; Zhu, Y. Antiviral activity of dandelion aqueous extract against pseudorabies virus both in vitro and in vivo. Front. Vet. Sci. 2023, 9.
Men, X.; Li, S.; Cai, X.; Fu, L.; Shao, Y.; Zhu, Y. Antiviral Activity of Luteolin against Pseudorabies Virus In Vitro and In Vivo. Animals. 2023, 13.
Point 5: Section 3.4: Figure 4A-B: missing in running text.
Response 5: We appreciate this commet very much. I've added the corresponding content in line 288 and 299.
Point 6: Figure 7: Author claims that there is a significant increase in TNF-alpha, IFN-gamma, and IL-6 levels. This sentence has to be redrafted correctly as there is moderate effect on these cytokine concentrations.
Response 6: Thank you very much for your valuable suggestions. The levels of TNF-α, IFN-γ, and IL-6 were all p < 0.05. It can be understood that the levels of TNF-α, IFN-γ and IL-6 in the serum of PRV infected mice were significantly higher than those in the control group.
We try our best to improve the manuscript and have made some changes marked in red in revised manuscript which will not influence the content and framework. We appreciate Editors and Reviewer’ warm work earnestly, and hope the revision will meet with approval. Once again, thank you very much for your comments and suggestions.
Kind regards,
Yan Zhu
Corresponding author.

Round 2
Reviewer 1 Report
Comments and Suggestions for Authors
The authors have addressed most of concerns.